# Targeted Self-Emulsifying Drug Delivery Systems to Restore Docetaxel Sensitivity in Resistant Tumors

**DOI:** 10.3390/pharmaceutics14020292

**Published:** 2022-01-26

**Authors:** Virginia Campani, Iris Chiara Salaroglio, Valeria Nele, Joanna Kopecka, Andreas Bernkop-Schnürch, Chiara Riganti, Giuseppe De Rosa

**Affiliations:** 1Department of Pharmacy, University of Naples Federico II, Via Domenico Montesano 49, 80131 Naples, Italy; virginia.campani@unina.it (V.C.); valeria.nele@unina.it (V.N.); 2Department of Oncology, University of Torino, Via Santena 5/bis, 10126 Torino, Italy; irischiara.salaroglio@unito.it (I.C.S.); joanna.kopecka@unito.it (J.K.); chiara.riganti@unito.it (C.R.); 3Department of Pharmaceutical Technology, Institute of Pharmacy, University of Innsbruck, Innrain 80/82, A-6020 Innsbruck, Austria; andreas.bernkop@uibk.ac.at

**Keywords:** docetaxel, self-emulsifying drug delivery system, parenteral administration, multidrug resistance, tumor targeting

## Abstract

The use of chemotherapeutic agents such as docetaxel (DTX) in anticancer therapy is often correlated to side effects and the occurrence of drug resistance, which substantially impair the efficacy of the drug. Here, we demonstrate that self-emulsifying drug delivery systems (SEDDS) coated with enoxaparin (Enox) are a promising strategy to deliver DTX in resistant tumors. DTX partition studies between the SEDDS pre-concentrate and the release medium (water) suggest that the drug is well retained within the SEDDS upon dilution in the release medium. All SEDDS formulations show droplets with a mean diameter between 110 and 145 nm following dilution in saline and negligible hemolytic activity; the droplet size remains unchanged upon sterilization. Enox-coated SEDDS containing DTX exhibit an enhanced inhibition of cell growth compared to the control on cells of different solid tumors characterized by high levels of FGFR, which is due to an increased DTX internalization mediated by Enox. Moreover, only Enox-coated SEDDS are able to restore the sensitivity to DTX in resistant cells expressing MRP1 and BCRP by inhibiting the activity of these two main efflux transporters for DTX. The efficacy and safety of these formulations is also confirmed in vivo in resistant non-small cell lung cancer xenografts.

## 1. Introduction

Despite the growing number of approved treatments and clinical trials, cancer is the second leading cause of death globally according to the WHO. To date, most of the available anticancer treatments involve the use of chemotherapeutic agents at high doses, which induce non-specific side effects; in addition, resistance to chemotherapy can develop over time, making the treatment ineffective [1]. Multidrug resistance (MDR) to different and structurally unrelated drugs is mainly due to alterations in drug kinetics, targets, and high expression of efflux transporters [2]. In particular, efflux drug transporters are present in a large number in drug-resistant cells and are involved in the efflux of anticancer drugs [3,4]. Docetaxel (DTX) is a water-insoluble, semisynthetic taxoid and is a well-established chemotherapeutic agent with a direct antitumoral activity due to the onset of an apoptotic cascade mediated by BCL-2 phosphorylation [5]. DTX is clinically used for the treatment of various tumors such as metastatic breast, lung, and prostate cancer [5,6,7,8]. When administered by intravenous infusion, DTX exhibits a linear pharmacokinetics with an extensive liver metabolization and patients with advanced cancer generally showed very high blood toxicity [5]. Furthermore, patients often become nonresponsive to DTX antitumoral therapy due to the onset of drug resistance mechanisms mediated by multidrug transporters such as the ATP-binding cassette (ABC) transporters [7,8,9]. More specifically, drug resistance to DTX is ascribed to the upregulation of P-glycoprotein (Pgp, ABCB1), MDR-related protein 1 (MRP1/ABCC1), and breast cancer resistance protein/ATP-binding cassette G2 (BCRP/ABCG2), which are the main efflux transporters for DTX [2]. As such, there is an unmet need for new therapies with minimal side effects and long-term efficacy.

Nanomedicines have been successfully proposed for targeted anticancer therapies and their ability to overcome MDR is under investigation [10,11]. However, there is a discrepancy between the number of nanomedicines currently used in clinical settings and the number of studies on the use of nanotechnology for cancer treatment. This may be due the technology transfer challenges to move from the bench to the bedside. In this context, self-emulsifying drug delivery systems (SEDDS) are an emerging technology for the systemic delivery of drugs since they are easy to set up, thermodynamically stable, and can encapsulate highly lipophilic drugs [12,13]. SEDDS comprise a mixture of lipids, surfactants, and co-solvents that spontaneously form an oil-in-water nanoemulsion, encapsulating the active compound upon dispersion in aqueous media [14]. While SEDDS are a well-established technology for the oral administration of poorly water-soluble drugs [15], their potential for drug delivery upon intravenous (i.v.) administration remains unexplored and has been only recently demonstrated by our group [16]. More in detail, we designed and optimized the formulation of SEDDS coated with enoxaparin (Enox), which showed enhanced cellular uptake compared to uncoated SEDDS in cancer cells [16]. Low molecular weight heparins (LMWH) such as enoxaparin (Enox) are natural glycosaminoglycans derived from the fractionation of heparin, which is the gold standard for the treatment of thrombosis [17,18,19]. Due to their negative charge, LMWH are able to bind a large amount of intracellular and extracellular matrix components, which are also involved in tumor progression and influence their activity [19,20]. It has been demonstrated that LMWH can contrast MDR of many anticancer drugs due to its ability to bind several drug transporters of the ABC and non-ABC families [19,21]. Heparin and its derivatives can interact with ABC drug transport proteins, directly inhibit ATPase activity, and reduce the efflux of chemotherapeutic agents, thus enhancing their cytotoxicity. Furthermore, heparin and LMWH have been recently shown to interact with lung resistance protein (LRP), the main non-ABC transport protein [19,21], and with vascular endothelial, fibroblast, and angiogenetic growth factor receptors, which are strictly involved in tumoral angiogenetic processes [22,23]. All this evidence suggests the potential of LMWH as ligands on delivery systems for targeting chemo-resistant cells.

In this work, we propose Enox-coated SEDDS as a novel platform for the systemic delivery of DTX able to overcome drug resistance towards DTX in cancer cells. We developed Enox-coated SEDDS containing DTX and carried out an extensive physico-chemical characterization to evaluate the solubility and distribution coefficient of DTX within SEDDS, the colloidal dimension and surface charge of Enox-coated SEDDS formulations, as well as their stability against aggregation in biological fluids. For the initial in vitro screening we focused on two types of tumors, namely breast cancer and non-small cell lung cancer, for which DTX and taxanes are used as first-line or second-line therapy, respectively [24,25]. After a preliminary screening of the SEDDS cytotoxic potential, we focused on A549 non-small cell lung cancer cells since these cells express high levels of multiple ABC transporters (Pgp, MRP1, and BCRP), thus exhibiting a strong resistance to DTX. We demonstrated that the efficacy of Enox-coated SEDDS formulations was due to enhanced cellular uptake mediated by fibroblast growth factor receptor 1 (FGFR1)-triggered endocytosis; cells silenced for FGFR1 did not show comparable SEDDS uptake. Additionally, the Enox coating could directly inhibit the catalytic cycle of MRP1 and BCRP. The combination of these two mechanisms increased the retention and the cytotoxic effect of DTX encapsulated within Enox-coated SEDDS. Finally, the anticancer activity of SEDDS was investigated in vivo in DTX-resistant A549 xenografts models that are refractory to the antitumor effect of free DTX. Consistent with the in vitro results, the Enox-coated SEDDS formulation encapsulating DTX was the most effective in reducing tumor growth without adding systemic toxicity. These promising results in terms of safety and efficacy may pave the way to the further development of our formulations towards clinical settings.

## 2. Materials and Methods

### 2.1. Materials and Cell Lines

Enoxaparin (Enox, average MW 4500 Da) was purchased from Sanofi-Aventis GmbH (Wien, Austria). PeceolTM (glyceryl monooleate) and Labrafil^®^ M 1944 (oleoyl polyoxyl-6 glycerides) were a gift from Gattefossé (Saint-Priest, France). Palmitoyl chloride (PC), Cremophor EL (polyoxyl-35 castor oil), propylene glycol (PG), Fe (III) chloride, L-cysteine ethyl ester hydrochloride, toluidine blue, sodium chloride (NaCl), Triton X-100, fluorescein diacetate (FDA), sodium hydroxide (NaOH), and tetrahydrofuran (THF) were obtained from Sigma-Aldrich Co. (Vienna, Austria). Docetaxel (DTX) was purchased by Enzo Life (Farmingdale, NY, USA), and sodium chloride, calcium chloride, sodium phosphate dibasic, potassium chloride, and bovine serum albumin (BSA) were obtained from Sigma-Aldrich Co. (Milan, Italy). Human plasma was obtained from healthy volunteers. All solvents, chemicals, and media were of analytical grade and used as received. Breast cancer MCF7, SKBR3, T74D, and MDA-MB-231 cells, and human non-small cell lung cancer (NSCLC) NCI-H1395, NCI-H1650, NCI-H1975, and A549 were provided from ATCC (Manassas, VA, USA), cultured in their respective media containing fetal bovine serum (10% *v*/*v*), penicillin-streptomycin (1% *v*/*v*), and L-glutamine (1% *v*/*v*). The non-targeting siRNA sequence (Trilencer-27 Universal scrambled negative control siRNA duplex, #R30004), the FGFR1-targeting siRNAs pool of 3 unique 27mer siRNA duplexes (#SR320159), the CRISPR pCas vectors targeting MRP1 (#KN418182), BCRP (#KN405640), or the non-targeting vector (#GE100003) were purchased from Origene (Rockville, MD, USA). Anti-FGFR1 antibody (#ab58516) and anti-MRP1/ABCC1 (#ab24102) were purchased from Abcam (Cambridge, UK), while the anti-Pgp/ABCB1 (15D3) and the anti-BCRP/ABCG2 (B1) were purchased from BD Biosciences (San Josè, CA, USA) and Santa Cruz Biotechnology Inc. (Santa Cruz, CA, USA), respectively. The anti-β-tubulin (D10) was also purchased from Santa Cruz Biotechnology Inc. and the secondary horseradish peroxidase-conjugated antibodies were obtained from Bio-Rad Laboratories (Hercules, CA, USA). Ki67 (AB9260) was purchased from Sigma Aldrich (Milan, Italy) while the peroxidase-conjugated secondary antibody was obtained from Dako (Glostrup, Denmark).

### 2.2. Methods

#### 2.2.1. Solubility Studies and Determination of Distribution Coefficient of Docetaxel (Log D_SEDDS/release medium_)

Solubility of DTX in SEDDS pre-concentrate was determined as previously reported by Griesser et al. [26] as well as by spectrophotometric analysis. Briefly, DTX at concentrations from 2 to 50 mg mL^−1^ was added to SEDDS pre-concentrate and samples were left under continuous stirring (1000 rpm, overnight) at room temperature. Then, samples were centrifugated (1000 rpm, 20 min), to separate any undissolved DTX, and drug solubility was evaluated. The highest amount of DTX dissolved in pre-concentrate SEDDS was considered its solubility. The amount of DTX soluble in pre-concentrate was determined after dilution of the samples in DMSO (1:100 *v*/*v*) and following UV analysis at λ = 268 nm. Finally, the Log D_(SEDDs/release medium)_ of DTX was determined as previously reported by Shahzadi et al. [27] with some modifications and according to Equation (1):(1)$$\mathrm{Log}\;D=\log\;\frac{\mathrm{Solubility}\;\mathrm{in}\;\mathrm{SEDDS}\;\mathrm{pre}-\mathrm{concentrate}}{\mathrm{Solubility}\;\mathrm{in}\;\mathrm{release}\;\mathrm{medium}}$$

#### 2.2.2. Synthesis of Enoxaparin-Palmitoyl Conjugate (Enox-Pa)

The Enox-Pa (Enox/PC 1:200 molar ratio) conjugate was synthesized as reported by Giarra et al. [16]. Briefly, an organic solution of PC in tetrahydrofuran and an aqueous solution of Enox (500 µg mL^−1^) were mixed (1:1 *v*/*v*) and incubated at 37 °C for 1 h under stirring. Thereafter, samples were shaken at 25 °C for about 16 h to allow for the evaporation of the organic phase. Finally, the aqueous solution containing Enox-Pa conjugate was centrifugated (13,000 rpm for 20 min) to remove the unconjugated PC and lyophilized. 

#### 2.2.3. Preparation and Characterization of SEDDS Coated with Enox-Pa and Loaded with DTX 

The SEDDS pre-concentrate was prepared by vortex mixing the excipients Peceol (20% *w*/*w*), Cremophor EL (40% *w*/*w*), Labrafil-1944 (30% *w*/*w*), and PG (10% *w*/*w*) at room temperature, as previously reported [16]. For the preparation of SEDDS coated with Enox-Pa and encapsulating DTX (SEDDS/DTX/Enox-Pa), the pre-concentrate was mixed to Enox-Pa conjugates dissolved in DMSO under magnetic stirring (700 rpm, 25 °C, overnight) at a final concentration of 1 mg of Enox-Pa/g SEDDS. The pre-concentrate SEDDS/Enox-Pa was then purified by dialysis (cut-off 16,000 Da) for 4 h against water. Afterwards, DTX powder was added to the SEDDS/Enox-Pa pre-concentrate (19.6 mg DTX mL^−1^) and left under magnetic stirring until complete dissolution of DTX. The complete dissolution of DTX was confirmed following sample centrifugation (1000 rpm for 20 min) and the analysis of the solution as reported above [26,28]. SEDDS coated with Enox-Pa and loaded with DTX were obtained by dispersing the pre-concentrate prepared as reported above in 20 mM buffer solution, pH 7.4 (1% *w*/*v* in PBS). Uncoated SEDDS and blank SEDDS coated with Enox-Pa were prepared similarly. All the SEDDS formulations were characterized in terms of mean diameter, PI, and ζ-potential of SEDDS droplets by Zetasizer Ultra (Malvern, UK). All measurements were performed in triplicate (n = 3) and data are the average of results carried out on at least three different batches.

#### 2.2.4. Stability Studies in Biological Media

##### Stability in Bovine Serum Albumin (BSA) Solution and Human Plasma

The stability of SEDDS formulations after direct interaction with serum proteins was assessed in the presence of bovine serum albumin (BSA) and then in human plasma. Thus, a BSA buffer solution (20 mM phosphate buffer pH 7.4) was used. In the case of stability studies in plasma, human plasma was separated from erythrocytes by centrifugation of human blood (2000 rpm for 15 min) and then diluted with 20 mM phosphate buffer, pH 7.4 (1% *v*/*v*). SEDDS formulations were incubated at 37 °C in both media (1% *w*/*v*) for up to 4 h. The interaction between SEDDS and serum proteins was evaluated by monitoring any variation of size and PI [29].

##### Hemolysis Assay

Hemolytic activity of SEDDS formulations was tested as reported in our previous work [27]. Briefly, erythrocytes were separated from plasma (centrifugation at 2000 rpm for 15 min) and resuspended with saline solution (NaCl 0,9% *w*/*v*) three times. Then, erythrocytes were diluted (1:10) with NaCl 0.9% *w*/*v* solution, and SEDDS formulations were incubated (0.2% *w*/*v*) for 4 h at 37 °C in a shaker bath. A 1:10 dilution of red cells with saline solution or erythrocytes in excess of water were used as the negative (0% hemolysis) and positive (100% hemolysis) control, respectively. Afterwards, samples were placed on ice for 2 min to quench erythrocyte lysis and centrifuged (3000 rpm for 5 min) to separate the supernatant from intact erythrocytes. The hemoglobin content in the supernatant was determined by spectrophotometer (Thermo Fisher Scientific 1510 Multiskan Go, Waltham, MA, USA) measuring the absorbance at λ = 540 nm.

#### 2.2.5. Sterilization of SEDDS

To check the possibility to use sterile formulations, SEDDS formulations (1% *w*/*v* in 0.9% *w*/*v* NaCl solution) were characterized in terms of size and PI before and after filtration trough 0.2 µm acetate cellulose filters (Sartorius). All experiments were performed in triplicate and the results are reported as the mean ± SD.

#### 2.2.6. In Vitro Studies

##### Cell Viability

Cells (1 × 10^5^) were seeded in 96-well plates and incubated for 72 h as indicated in the Results Section. Cell viability was measured with the ATPLite kit (PerkinElmer, Waltham, MA, USA), as per the manufacturer’s instructions, using a Synergy HT Multi-Detection Microplate Reader (Bio-Tek Instruments, Winooski, VT, USA). The relative luminescence units (RLUs) of the untreated cells (ctrl) were considered 100%; the RLUs of the other experimental conditions were expressed as percentage versus control cells. 

##### Cell Silencing and Knock-Out

For the transient silencing of fibroblast growth factor receptor 1 (FGFR1), cells (1 × 10^5^) were treated with a non-targeting siRNA sequence (Trilencer-27 Universal scrambled negative control siRNA duplex) or with a FGFR1-targeting siRNAs pool of 3 unique 27mer siRNA duplexes, according to the manufacturer’s instructions. To generate the knocked-out clones for MRP1/ABCC1 or BCRP/ABCG2, cells (5 × 10^5^) were transduced with CRISPR pCas vectors targeting MRP1, BCRP (1 µg), or a non-targeting vector (1 µg), according to the manufacturer’s instructions. Stable knocked-out cells were selected by culturing cells in the presence of puromycin (1 µg mL^−1^) for 4 weeks. The levels of silenced or knocked-out proteins were verified by immunoblotting. 

##### Immunoblotting

Cells were rinsed in lysis buffer (0.4 mL, 125 mM Tris-HCl, 750 mM NaCl, 1% *v*/*v* NP40, 10% *v*/*v* glycerol, 50 mM MgCl_2_, 5 mM EDTA, 25 mM NaF, 1 mM NaVO_4_, 10 μg mL^−1^ leupeptin, 10 μg mL^−1^ pepstatin, 10 μg mL^−1^ aprotinin, 1 mM phenylmethylsulfonyl fluoride, pH 7.5), sonicated, and centrifuged at 13,000× *g* for 10 min at 4 °C. Proteins (25 μg) were subjected to SDS-PAGE and probed with the following antibodies: anti-FGFR1 antibody (1:500), anti-Pgp/ABCB1 (1:250), anti-MRP1/ABCC1 (1:250), anti-BCRP/ABCG2 (1:500), and anti-β-tubulin (1:500), followed by secondary horseradish peroxidase-conjugated antibodies (Bio-Rad Laboratories, Hercules, CA, USA). The proteins were detected by enhanced chemiluminescence (Bio-Rad Laboratories).

##### SEDDS Uptake

Cells were incubated with the respective formulations of SEDDS, labelled with FDA (0.1%), for 1, 3 and 6, and 24 h, then rinsed with PBS twice, detached by gentle scraping, sonicated, and re-suspended in PBS (300 µL). The amount of fluorescence in the cell lysates was measured by spectrophotometric analysis using a Synergy HT Multi-Detection Microplate Reader (Bio-Tek Instruments). Excitation and emission wavelengths were 475 and 520 nm, respectively. A blank with cells without fluorescently labelled SEDDS was prepared in each set of experiments, and its fluorescence was subtracted. The relative fluorescence units (RFUs) measured in the solution of fluorescently labelled SEDDS before incubation (t0 fluorescence) was considered as 100% fluorescence. Intracellular RFUs, considered an index of uptake, were expressed as % of fluorescence versus t0 fluorescence. In fluorescence microscopy analysis, cells (0.5 × 10^5^) were grown on sterile glass coverslips, treated with fluorescently labelled SEDDS for 6 h, rinsed with PBS, fixed with paraformaldehyde (4% *w*/*v*) for 15 min, washed three times with PBS and incubated with 4′,6-diamidino-2-phenylindole dihydrochloride (DAPI, diluted 1:20,000) for 3 min at room temperature in the dark, then washed three times with PBS and once with water. The slides were mounted with of Gel Mount Aqueous Mounting (4 μL) and examined using a Leica DC100 microscope (Leica Microsystems, Wetzlar, Germany) with a 63× oil immersion objective and 10× ocular lens.

##### ATPases Activity

The ATPase activity of Pgp, MRP1, and BCRP, taken as an index of the transporters catalytic activity, was measured after washing cells with Ringer’s solution (148.7 mM NaCl, 2.55 mM K_2_HPO_4_, 0.45 mM KH_2_PO_4_, 1.2 mM MgSO_4_; pH 7.4), and in lysis buffer (10 mM Hepes/Tris, 5 mM EDTA, 5 mM EGTA, 2 mM dithiothreitol; pH 7.4) supplemented with phenylmethylsulfonyl fluoride (2 mM), aprotinin (1 mM), pepstatin (10 μg mL^−1^), and leupeptin (10 μg mL^−1^). Lysates were centrifuged at 300× *g* for 10 min in the pre-centrifugation buffer (10 mM Tris/HCl, 25 mM sucrose; pH 7.5), overlaid on a sucrose cushion (10 mM Tris/HCl, 35% *w*/*v* sucrose, 1 mM EDTA; pH 7.5) and then centrifuged at 14,000× *g* for 10 min. The interface was collected, diluted in the centrifugation buffer (5 mL, 10 mM Tris/HCl, 250 mM sucrose; pH 7.5), and centrifuged at 100,000× *g* for 45 min to collect the membrane-enriched fraction. The pellet was resuspended in the centrifugation buffer (0.5 mL), sonicated, and an aliquot (100 µL) was used for protein quantification. Proteins (50 0 μg) were immunoprecipitated overnight at 4 °C with the anti-Pgp, anti-MRP1, or anti-BCRP antibodies. The immunopurified Pgp, MRP1, or BCRP (20 µg) were incubated for 30 min at 37 °C with the reaction mix (50 μL, 25 mM Tris/HCl, 3 mM ATP, 50 mM KCl, 2.5 mM MgSO_4_, 3 mM dithiothreitol, 0.5 mM EGTA, 2 mM ouabain, 3 mM NaN_3_; pH 7.0). The reaction was stopped by adding ice-cold stopping buffer (0.2 mL, 0.2% *w*/*v* ammonium molybdate, 1.3% *v*/*v* H_2_SO_4_, 0.9% *w*/*v* SDS, 2.3% *w*/*v* trichloroacetic acid, 1% *w*/*v* ascorbic acid). After 30 min incubation at room temperature, the absorbance of the phosphate hydrolyzed from ATP, taken as index of the catalytic activity of the transporters, was measured at 620 nm using a Packard EL340 microplate reader (Bio-Tek Intruments). The absorbance was converted into nanomoles hydrolyzed phosphate (Pi)/min/mg proteins, according to the titration curve previously prepared.

#### 2.2.7. In Vivo Experiments

In the experiments, 6-week-old Balb/C female nude mice were subcutaneously injected with A549 cells (1 × 10^6^) in Matrigel (100 µL). When tumors reached the volume of 100 mm^3^, mice (n = 8/group) were randomized in the following groups and treated once a week for 6 weeks as reported: (1) the vehicle group, treated with Intralipid intravenously (100 µL, i.v.); (2) DTX 2.5 mg kg^−1^, dissolved in Intralipid (100 µL, i.v.); (3) DTX 5 mg kg^−1^, dissolved in Intralipid (100 µL, i.v.); (4) SEDDS containing DTX at 2.5 mg kg^−1^ final concentration, diluted in saline (100 µL, i.v.); (5) SEDDS containing DTX at 5 mg kg^−1^ final concentration, diluted in saline (100 µL i.v.); (6) Enox-coated SEDDS containing DTX at 2.5 mg kg^−1^ final concentration, diluted in saline (100 µL, i.v.); (7) Enox-coated SEDDS containing DTX at 5 mg kg^−1^ final concentration, diluted in saline (100 µL, i.v.). Animals were euthanized at week 7 with zolazepam (0.2 mL kg^−1^) and xylazine (16 mg kg^−1^) injected intramuscularly. Tumors were excised, photographed, fixed in paraformaldehyde (4% *v*/*v*) overnight, and embedded in paraffin. The paraffin sections were stained with hematoxylin/eosin or immunostained for Ki67 (1:100), as an index of cell proliferation, followed by a peroxidase-conjugated secondary antibody (1:100). Liver, kidneys, and spleen were excised, fixed, and the paraffin sections were examined after hematoxylin/eosin staining. The sections were examined with a Leica DC100 microscope (Leica). At 3.5 weeks and immediately after the euthanasia, blood (200 µL) was collected to measure the following parameters: red blood cells (RBC), white blood cells (WBC), hemoglobin (Hb), and platelets (PLT), as indexes of bone marrow function; lactate dehydrogenase (LDH), aspartate aminotransferase (AST), alanine aminotransferase (ALT), and alkaline phosphatase (AP), as indexes of liver function; creatinine, as an index of kidney function; and creatine phosphokinase (CPK) as an index of muscle/heart damage, using commercially available kits from Beckman Coulter Inc. (Miami, FL, USA). Animal care and experimental procedures were approved by the Bio-Ethical Committee of the Italian Ministry of Health (#627/2018-PR).

#### 2.2.8. Statistical Analysis

All data in the text and figures are provided as means ± SEM. The results were analyzed by a one-way analysis of variance (ANOVA) and Tukey’s test. *p* <0.05 was considered significant.

## 3. Results and Discussion

Self-emulsifying drug delivery systems (SEDDS) are proposed here as a novel DTX delivery system to overcome common issues associated to the use of DTX in therapy, such as the low DTX solubility in water, which requires the use of toxic co-solvent, the poor selectivity among cancer and healthy cells, and the chemoresistance occurring in some tumors. SEDDS are a well-established technology to deliver poorly water-soluble drugs by the oral route [15]. Recently, our group proposed the use of SEDDS for the intravenous administration of lipophilic drugs; in particular, we showed that SEDDS can be modified with enoxaparin for cancer cell targeting [16]. Here, we intend to demonstrate that enoxaparin-modified SEDDS can represent a powerful strategy to deliver DTX in chemo-resistant tumors; we also aim at providing insights into the mechanisms of cell targeting and evasion of drug resistance. 

The first step of the study was the evaluation of drug loading and solubility in SEDDS. The partition coefficient of DTX between the SEDDS pre-concentrate and the release medium was calculated; this step is crucial when designing SEDDS-based formulations for the systemic delivery of drugs to prevent the rapid drug release upon dilution in biological fluids. From a general point of view, drug loading in SEDDS corresponds to the drug solubility into the pre-concentrate. Thus, in the first step of the study, the drug loading in SEDDS pre-concentrate was determined as previously reported [26,30]. Undissolved DTX was observed at concentrations higher than 20.5 mg mL^−1^ and spectrophotometric analysis confirmed that DTX is soluble in the SEDDS pre-concentrate at a concentration of 19.67 ± 0.08 mg mL^−1^. Therefore, DTX/SEDDS pre-concentrate at a DTX concentration of 19.60 mg mL^−1^ was used throughout the study. Drug release from SEDDS should be mainly due to drug diffusion towards the external aqueous phase, and the migration through the interfacial barrier of the system may be neglected [30]. Hence, drug release from SEDDS should be reasonably influenced by the solubility of the drug in the release phase [28,30]. Based on these considerations, the drug partition coefficient between the SEDDS pre-concentrate and the release medium, i.e., log D SEDDS/release medium, can be regarded as predictive of drug release from SEDDS following i.v. administration [28]. By using 0.274 mg L^−1^ as the DTX aqueous solubility [31], a value of 4.80 ± 0.01 was obtained for the log D SEDDS/release medium. Bernkop-Schnürch and Jalil [30] reported that a Log D lower than 3 can lead to a significant amount of drug immediately released from the SEDDS droplets, while a Log D > 5 results in the extensive retention of the lipophilic drug within the oily droplets upon dilution in biological media [30]. In this study, the Nernstschen distribution equation [30] was used to predict DTX release from SEDDS. Since 1 mL of SEDDS is injected in a blood volume of approximately 6 L, we hypothesized that the extent of drug release in the medium should be negligible and approximately 99.9% of the encapsulated DTX should remain within the SEDDS. These data support the use of SEDDS for the delivery of DTX. Despite its well-established clinical use, the low water solubility of DTX requires the use of cosolvents (e.g., ethanol) for its administration. The low selectivity for tumor cells and the need for organic solvents elicit many undesirable side effects and foster the development of novel formulations, including nanotechnology-based approaches [32]. In this context, SEDDS could represent a novel formulation able to avoid the use of organic solvents and easy to scale up. Moreover, the modification of SEDDS with Enox could endow SEDDS with the ability to target tumor cells and overcome tumor cell resistance to DTX. Heparin and LMWH derivatives (e.g., Enox) can target different receptors and substrates within the tumor microenvironment. In particular, heparin and its derivatives interact with FGFR and with drug transporters involved in drug efflux and MDR [21,33]. Notably, it has been reported that heparin is able to inhibit the function of ABC transporters by reducing the efflux rate of chemotherapeutic drugs from cancer cells. Indeed, heparin and its derivatives also reduced ATPase activity of some transporters at very low concentrations [21,33]. Thus, the use of heparin and its derivatives has been proposed to enhance the toxicity and the antitumor activity of different anticancer drugs in MDR cells [21,33]. In our previous study, we developed Enox-coated SEDDS, and we demonstrated that Enox led to an increase of internalization efficiency in two different types of adenocarcinoma cell lines [16].

The amphiphilic conjugate (Enox-Pa) used to functionalize the SEDDS surface was obtained by the covalent binding of the fatty acid palmitoyl chloride (PC) with the hydroxyl groups of Enox. Based on our previous study, the Enox-Pa conjugate was prepared at an E/PC molar ratio of 1:200, which ensures an optimal degree of reaction, with about 84.38% of hydroxyl groups of enoxaparin substituted and conjugated to PC [16]. Then, SEDDS targeted with Enox-Pa (S/Enox-Pa) were prepared by mixing the SEDDS pre-concentrate with Enox-Pa (1:1 *w*/*w*) at room temperature. DTX was added to S/Enox-Pa under stirring, obtaining S/Enox-Pa/DTX. SEDDS pre-concentrates were diluted in 20 mM phosphate buffer solution at pH 7.4 to obtain a clear emulsion. The resulting SEDDS were then characterized in terms of droplet size, polydispersity index (PI), and zeta potential (Table 1). Plain SEDDS had a mean diameter of about 110 nm; the mean size of droplets slightly increased following DTX addition (S/DTX) or after the inclusion of Enox-Pa (S/Enox-Pa). Instead, SEDDS encapsulating DTX and targeted with Enox-Pa (S/Enox-Pa/DTX) had a mean diameter of about 145 nm. All the SEDDS formulations were characterized by a homogeneous distribution of droplets size with a PI value around 0.2. SEDDS showed a negative zeta potential that further decreased when the negatively charged enoxaparin was included in the formulation, thus suggesting the exposure of Enox on the droplet surface.

Studies were performed to predict possible interactions between SEDDS formulations and serum components following i.v. administration. The mean diameter and the PI of the different SEDDS formulations were monitored following incubation in albumin and human plasma at 37 °C for up to 4 h (Table 2). Physical alterations due to droplet aggregation or protein adsorption on the surface of SEDDS could be indicative of poor hemocompatibility of SEDDS [34,35]. These events might hamper the i.v. administration of SEDDS, leading to blood vessel occlusion and rapid elimination from the systemic circulation due to the capture by macrophages [36,37,38]. As shown in Table 2, all the formulations exhibited good stability against aggregation following incubation in BSA solution and human plasma, where no changes in size and PI were observed.

Thereafter, a hemolysis assay was carried out on the different SEDDS formulations; this test is considered predictive of the level of damage and lysis of erythrocytes cytoplasmic membrane following i.v. administration. The hemolytic activity was evaluated according to ASTM F 756-17 [29]. The results (reported in Table 3) showed that all the formulations were characterized by a very low hemolytic activity (<5%). It is worthy to note that SEDDS loaded with DTX and coated with Enox showed a hemolytic percentage < 2%, which is regarded as not hemolytic.

In the perspective of a parenteral administration, sterility of formulations is a mandatory precondition. Thus, we verified the possibility to sterilize the formulations by filtration to avoid high temperature sterilization. All SEDDS formulations were diluted in saline solution (0.9% *w*/*v* NaCl), and the mean diameter and PI were measured before and after filtration with 0.22 μm acetate cellulose filters. As reported in Table 4, SEDDS diameter and PI did not significantly change after filtration.

To evaluate the biological efficacy of SEDDS with DTX and Enox-Pa, we analyzed a panel of human cell lines from breast and human non-small cell lung cancer (NSCLC) that are treated with DTX as first- or second-line treatment options [25,39]. The cell lines had different levels of FGFR1, a putative interactor for heparin [40], and of ABC transporters involved in docetaxel efflux, i.e., Pgp, MRP1, and BCRP [2], as confirmed by immunoblotting (Appendix A). Cells were treated with SEDDS formulations at a concentration of 0.25% *w*/*v*, which ensures good stability in serum and plasma and no hemolytic activity [16]. In this experimental condition, the concentration of encapsulated DTX was 80 µM; therefore, we compared the effect of free DTX and DTX loaded in SEDDS on cell viability at this concentration. MCF7, SKBR3, and T47D breast cancer cells with low expression of the ABC transporters were sensitive to DTX (Figure 1a). DTX-loaded SEDDS with (S/Enox-Pa/DTX) or without (S/DTX) Enox coating were not superior to DTX administered as free drug (Figure 1a). The same profile was observed in NSCLC NCH-H1395 cells, which expressed low levels of ABC transporters, except for BCRP (Figure 1b). By contrast, Pgp/MRP1/BCRP-expressing cells, such as breast cancer MDA-MB-231 cells, NSCLC NCI-H1650, NCI-H1975, and A549 cells were resistant to free DTX, but they were significantly killed by both S/DTX and S/Enox-Pa/DTX. For all the cell lines tested, S/Enox-Pa/DTX was the most potent formulation compared to S/DTX and free DTX. Interestingly, in DTX-resistant cells, the efficacy of S/Enox-Pa/DTX was comparable to that of the association of nintedanib, a FGFR inhibitor, and DTX (Figure 1), a combination therapy used in chemo-refractory breast cancer and NSCLC as a second-line treatment [41,42]. Therefore, S/Enox-Pa/DTX may have a translational perspective against these tumors, which are resistant to the first line of treatment. Blank SEDDS, with or without Enox coating (S and S/Enox-Pa), did not elicit any decrease in cell viability (Figure 1), which excluded SEDDS-related cytotoxicity.

Since the maximal benefit of S/Enox-Pa/DTX was achieved in cells expressing high levels of FGFR1 and efflux transporters, we investigated whether the increased efficacy of S/Enox-Pa/DTX was due to a higher uptake mediated by FGFR1 and/or to a reduced efflux of DTX via ABC transporters. We focused on A549 cells that displayed the highest levels of FGFR1, Pgp, and MRP1 (Appendix A), and the highest resistance to free DTX (Figure 1b).

To evaluate whether the uptake of S/Enox-Pa/DTX was mediated by FGFR1, we silenced this receptor in A549 cells (Figure 2a), and we measured the intracellular uptake of fluorescently labelled SEDDS formulations. As shown in Figure 2b, the intracellular retention of SEDDS without E/Pa (S and S/DTX) did not change at all the time points analyzed. By contrast, S/Enox-Pa and S/Enox-Pa/DTX showed a time-dependent increase in accumulation within A549 cells (Appendix A and Figure 2b). At 1 h, we did not detect any fluorescence signal likely because the fluorescence was below the limit of detection of the microscope (Appendix A); the percentage of fluorophore taken up by the cells was below 25% using the most sensitive fluorometric quantification (Figure 2b). The fluorescence was more evident after 3 and 6 h following incubation with SEDDS; at 6 h, most cells were labelled, and the fluorescent signal was homogeneously distributed within the cytosol (Appendix A). The uptake of S/Enox-Pa and S/Enox-Pa/DTX within A549 cells was abrogated after FGFR1 silencing (Figure 2b,c and Appendix A), likely due to the lack of FGR1-triggered endocytosis of Enox-coated formulations. In agreement with these findings, S/DTX reduced cell viability in scrambled and siFGFR1 cells at the same extent, while the toxicity of S/Enox-Pa/DTX was lost in cells silenced for FGFR1 (Figure 2d). This experimental set suggests that FGFR1 is required for the intracellular delivery of DTX encapsulated within SEDDS coated with Enox-Pa. The highest cytotoxicity of S/Enox-Pa/DTX was not due to the inhibition of pro-survival pathways downstream FGFR1 [43] because untreated A549 cells and A549 cells treated with S/DTX or S/Enox-Pa/DTX had the same activity of the FGFR1 effectors Ras, ERK1/2, and Akt (Appendix A).

To explore whether the efficacy of S/Enox-Pa/DTX in cells resistant to the free drug could be also due to a decreased efflux of DTX via ABC transporters, we measured the catalytic efficiency of Pgp, MRP1, and BCRP extracted from A549 cells treated with the different SEDDS formulations (Figure 3a). As expected, blank SEDDS did not change the ATPase activity of these transporters. DTX, either as free drug or loaded within SEDDS, did not modify the transporters’ activity. Interestingly, the presence of Enox-Pa, in both S-Enox/Pa or S/Enox-Pa/DTX, significantly reduced the ATPase rate of MRP1 and BCRP, without affecting Pgp (Figure 3a). These results are in line with previous findings indicating that heparin reduces the activity of MRP1 and BCRP, increasing the intracellular retention of chemotherapeutic drugs substrates of these transporters such as doxorubicin, epirubicin, tamoxifen, and mitoxantrone [33]. To confirm our hypothesis, we produced A549 clones knocked-out for MRP1 and BCRP (Figure 3b), and we re-assessed the cytotoxicity elicited by DTX and the different SEDDS formulations. As expected, DTX and S/DTX significantly reduced the viability in A549 cells knocked-out for MRP1 (Figure 3c) or BCRP (Figure 3d). By contrast, S/Enox-Pa/DTX did not confer any further advantage compared to free DTX or S/DTX (Figure 3c,d). The same trend was obtained in A549 cells treated with the MRP1 inhibitor MK571 (Appendix A) or with the BCRP inhibitor fumitremorgin (Appendix A) at concentrations that were previously demonstrated to inhibit the activity of these transporters [44]. These data indicate that Enox-Pa reduces the activity of at least two transporters—MRP1 and BCRP—involved in DTX efflux. Together with the increased uptake via FGFR1, the reduced efflux via ABC transporters may explain the increased cytotoxicity of S/Enox-Pa/DTX in resistant cells.

Finally, we validated the anti-tumor efficacy and the safety profile of S/Enox-Pa/DTX in A549 xenografts. Mice were treated with two different dosages of free DTX—2.5 and 5 mg kg^−1^—which have been reported to be moderately effective in this experimental model [45]. Free DTX reduced tumor growth in a dose-dependent manner, but these results were not significantly different from the control (animals treated with vehicle, Figure 4a). S/DTX and S/Enox-Pa/DTX—at the same dosage of DTX—also elicited a dose-dependent reduction of the tumor growth rate, which was more pronounced than the free drug. S/Enox-Pa/DTX were significantly more effective than DTX and S/DTX in terms of tumor growth rate (Figure 4a) and tumor volume (Figure 4b,c). In agreement with the decreased proliferation observed in vitro, the intratumor proliferation, measured as positivity to Ki67, was reduced in this order: DTX<S/DTX<S/Enox-Pa/DTX (Figure 4d).

Importantly, neither S/DTX nor S/Enox-Pa/DTX altered the animal weight during the whole treatment (Appendix A) or showed signs of bone marrow, liver, kidney, and muscle toxicity at the mid-point of the study (Appendix A), according to the hematochemical parameters measured. DTX or S/DTX at the highest dosage reduced red blood cell, white blood cell, and platelet count at the end of the study. However, this toxicity was not produced by S/Enox-Pa/DTX (Appendix A), which was safer than free DTX or S/DTX. Consistently, no histological alterations were detected in the liver, kidney, and spleen analyzed post-mortem in animals treated with the highest dosage of S/Enox-Pa/DTX compared with animals treated with the vehicle (Appendix A). The lower toxicity of S/Enox-Pa/DTX can be explained by a more favorable tumor-to-normal tissues distribution that allows for an active targeting of S/Enox-Pa/DTX within the tumor site, where it can release DTX at effective anti-tumor concentrations. At the same time, non-transformed tissues may be spared from the undesired toxicities of DTX, reducing the undesired side effects.

## 4. Conclusions

This study provides elements to support the use of SEDDS for the delivery in DTX in tumors. DTX solubilization and DTX partition coefficient supported its loading in SEDDS without the risk of premature or incomplete release from the SEDDS droplets. Moreover, Enox-Pa-coated SEDDS containing DTX were not hemolytic and maintained their physico-chemical characteristics upon sterilization by filtration. Cells characterized by high levels of FGFR1 showed an enhanced inhibition of cell growth by using Enox-Pa-coated SEDDS containing DTX, likely favoring an increased internalization of DTX, as demonstrated in FGFR1-silenced cells. Only Enox-Pa-coated SEDDS were able to restore the sensitivity to DTX in cells expressing MRP1 and BCRP, the two main efflux transporters for DTX, and inducing resistance to this drug. This mechanism was due to the inhibition of MRP1 and BCRP activity. The combination of increased uptake and reduced efflux resulted in a substantial increase in the DTX intracellular retention and cytotoxicity. In vitro findings were confirmed in vivo, where Enox-Pa-coated SEDDS rescued the efficacy of DTX in resistant NSCLC xenografts. Finally, DTX loaded into Enox-Pa-coated SEDDS was significantly safer compared to free DTX, which may be ascribed to the accumulation of DTX at the tumor site. This study supports the further development of SEDDS for the i.v. administration of chemotherapeutics. We demonstrated that SEDDS can be an interesting alternative to currently used formulations because they are very easy to prepare (rapid scale up), can be targeted towards cells overexpressing FGFR1, which is commonly upregulated in cancer cells, and are able to restore DTX sensitivity in chemo-resistant tumors and to reduce the DTX systemic toxicity.

## Figures and Tables

**Figure 1 pharmaceutics-14-00292-f001:**
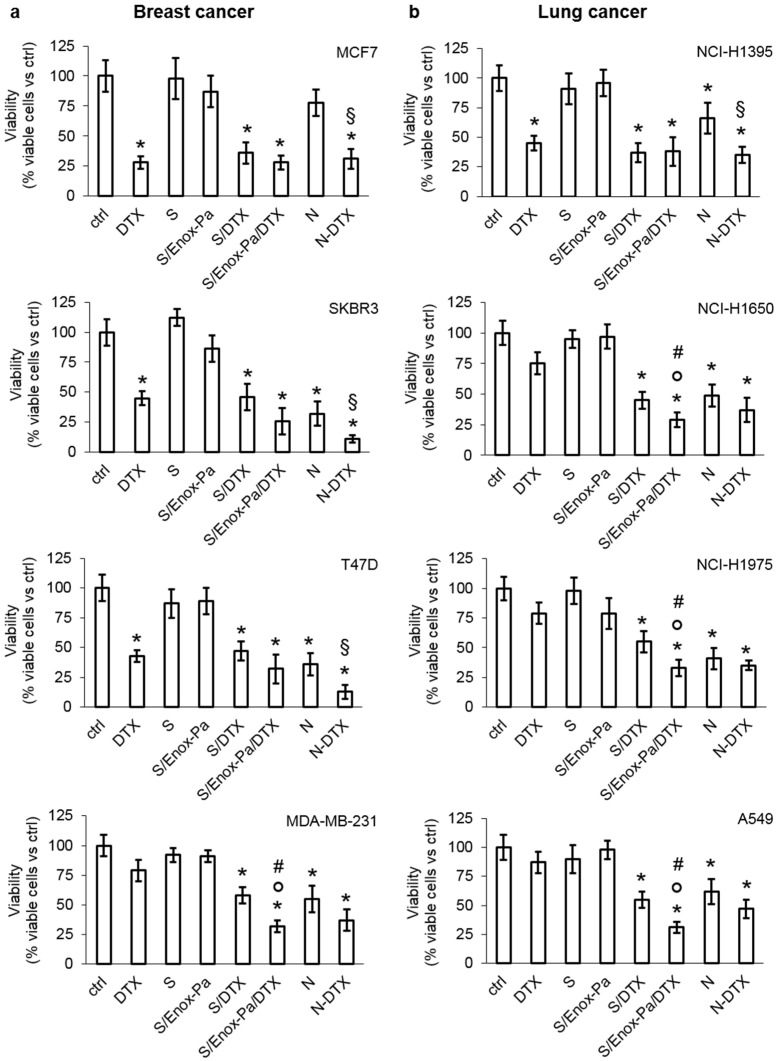
**Viability of breast and non-small cell lung cancer cells treated with free docetaxel and different formulations of SEDDs.** (**a**) Human breast MCF7, SKBR3, T47D, and MDA-MB-231 cells and (**b**) human non-small cell lung cancer NCI-H1395, NCI-H1650, NCI-H1975, and A549 cells were incubated for 72 h with fresh medium (ctrl), 80 µM free DTX, 0.25% *v*/*v* blank SEDDs (S), Enox-coated SEDDS (S/Enox-Pa), SEDDS containing DTX (80 µM final concentration; S/DTX), and Enox-coated SEDDS containing DTX (80 µM final concentration; S/Enox-Pa/DTX). Nintedanib (1 µM; N), alone or co-incubated with 80 µM DTX (N + DTX), was included as an inhibitor of FGFR1. Cell viability was measured by a chemiluminescence-based assay in quadruplicates. Data are presented as means ± SD (n = 3). * *p* < 0.001: vs. ctrl; ° *p* < 0.001: S/Enox-Pa/DTX vs. DTX; # *p* < 0.01: S/Enox-Pa/DTX vs. S/DTX; § *p* < 0.01: N + DTX vs. N.

**Figure 2 pharmaceutics-14-00292-f002:**
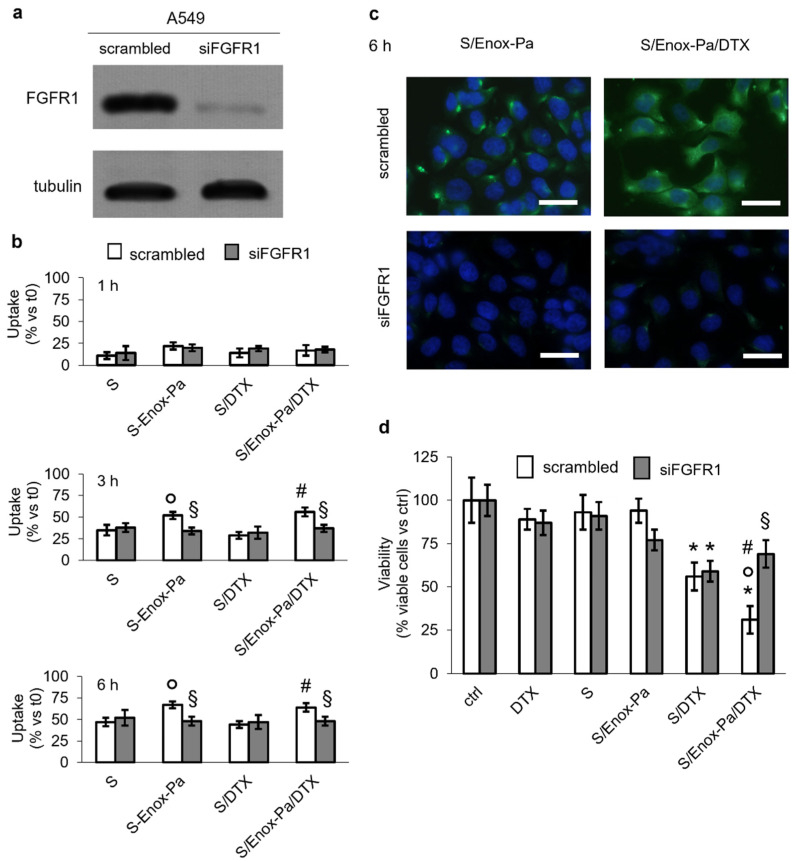
**FGFR1 mediates uptake and cytotoxicity of Enox-coated SEDDS containing DTX.** A549 cells were treated for 48 h with a non-targeting siRNA (scrambled) or with a pool of three siRNAs targeting FGFR1 (siFGFR1), then subjected to the following investigations. (**a**) Immunoblot of the indicated proteins. Tubulin was used as control of equal protein loading. The image is representative of one out of three experiments. (**b**) Cells were incubated for 1, 3, or 6 h with 0.25% *v*/*v* blank SEDDS (S), Enox-coated SEDDS (S/Enox-Pa), SEDDS containing DTX (80 µM final concentration; S/DTX), Enox-coated SEDDS containing DTX (80 µM final concentration; S/Enox-Pa/DTX). The intracellular fluorescence was measured in duplicates and compared to the fluorescence of the solution of each SEEDs before incubation (t0). The results, expressed as % of intracellular fluorescence versus fluorescence at t0, are presented as means ± SD (n = 3). ° *p* < 0.01: S/Enox-Pa vs. S; # *p* < 0.01: S/Enox-Pa/DTX vs. S/DTX; § *p* < 0.05: siFGFR1 vs. scrambled cells. (**c**) Representative photographs of scrambled and siFGFR1 A549 cells treated for 6 h with S-E and S-E-d, as in (**b**). The photos are representative of one out of three experiments. For each experimental condition, a minimum of five fields were examined. The photos are representative of one out of three experiments. Ocular: 10×; objective: 60×. Scale bar: 50 µm. (**d**) Cells were incubated with fresh medium (ctrl), 80 µM free docetaxel (DTX), 0.25% *v*/*v* blank SEDDs (S), Enox-coated SEDDS (S/Enox-Pa), SEDDS containing docetaxel (80 µM final concentration; S/DTX), and Enox-coated SEDDs containing docetaxel (80 µM final concentration; S/Enox-Pa/DTX) for 72 h. Cell viability was measured by a chemiluminescence-based assay in quadruplicates. Data are presented as means ± SD (n = 3). * *p* < 0.01: vs. ctrl; ° *p* < 0.01: S/Enox-Pa/DTX vs. DTX; # *p* < 0.001: S/Enox-Pa/DTX vs. S/DTX; § *p* < 0.001: siFGFR1 vs. scrambled cells.

**Figure 3 pharmaceutics-14-00292-f003:**
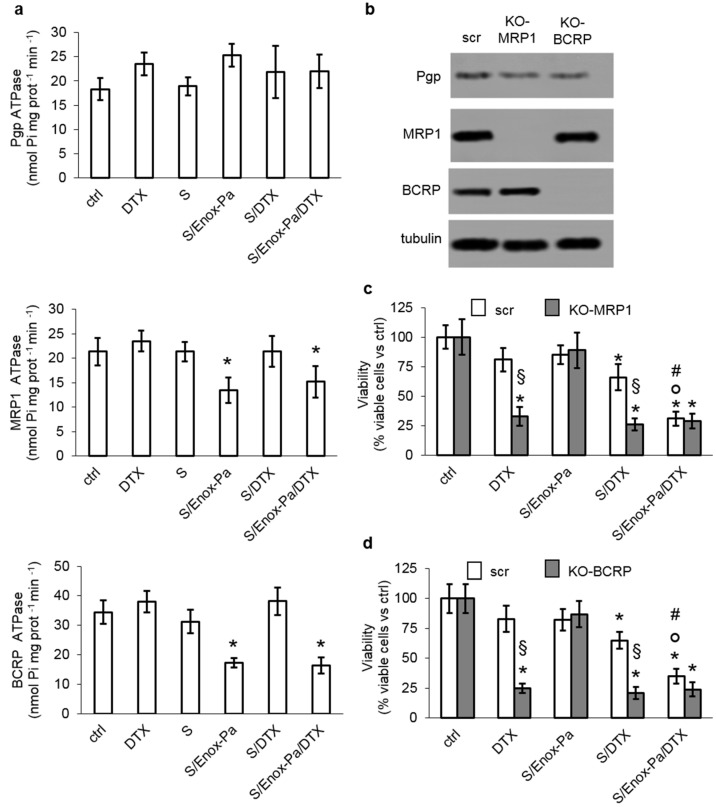
**Enox-coated SEDDS inhibit MRP1 and BCRP activity.** A459 cells were incubated for (**a**) 24 h or (**c**) 72 h with fresh medium (ctrl), 80 µM free docetaxel (DTX), 0.25% *v*/*v* blank SEDDS (S), Enox-coated SEDDS (S/Enox-Pa), SEDDs containing docetaxel (80 µM final concentration; S/DTX), and Enox-coated SEDDS containing docetaxel (80 µM final concentration; S/Enox-Pa/DTX). (**a**) The rate of ATP hydrolysis by immunopurified Pgp, MRP1, or BCRP extracted from cells treated as reported above was measured by spectrophotometric analysis in triplicates. Data are presented as means + SD (n = 3). * *p* < 0.02: vs. ctrl. (**b**) A549 cells were transduced with a non-targeting (scrambled) CRISPR-Cas vector or with a CRISPR-Cas vector to knock-out (KO) MRP1 or BCRP. The indicated proteins were measured by immunoblotting. Tubulin was used as control of equal protein loading. The image is representative of one out of three experiments. (**c**,**d**) The viability of scrambled, KO MRP1, and KO BCRP A549 cells was measured by a chemiluminescence-based assay in quadruplicates. Data are presented as means + SD (n = 3). * *p* < 0.001: vs. ctrl; ° *p* < 0.001: S/Enox-Pa/DTX vs. DTX; # *p* < 0.01: S/Enox-Pa/DTX vs. S/DTX; § *p* < 0.001: KO vs. scrambled cells.

**Figure 4 pharmaceutics-14-00292-f004:**
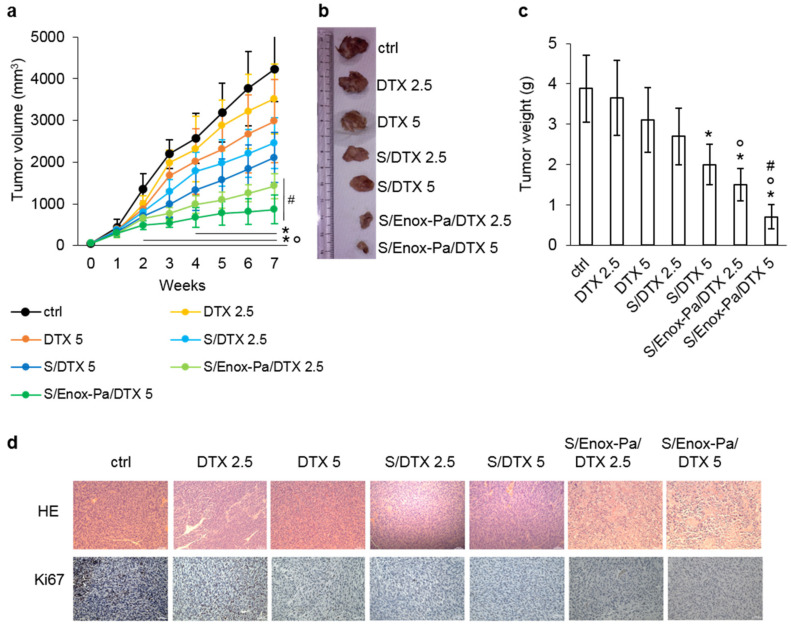
**Enox-coated SEDDS reduce the growth of drug resistant non-small cell lung cancer xenografts.** First, 1 × 10^6^ A549 cells were inoculated subcutaneously in the right flank of 6-week-old Balb/C female nude mice. When tumors reached the volume of 100 mm^3^, mice (n = 8/group) were randomized in the following groups and treated once a week for 6 weeks as reported: (1) vehicle group (ctrl), with 100 µL saline solution administered intravenously; (2) docetaxel 2.5 mg kg^−1^ (DTX 2.5) in 100 µL Intralipid i.v.; (3) docetaxel 5 mg kg^−1^ (DTX 5) in 100 µL Intralipid i.v.; (4) SEDDs containing docetaxel at 2.5 mg kg^−1^ final concentration (S/DTX 2.5) i.v.; (5) SEDDs containing docetaxel at 5 mg kg^−1^ final concentration (S/DTX 5) i.v.; (6) Enox-coated SEDDS containing docetaxel at 2.5 mg kg^−1^ final concentration (S/Enox-Pa/DTX 2.5) i.v.; (7) Enox-coated SEDDS containing docetaxel at 5 mg kg^−1^ final concentration (S/Enox-Pa/DTX 5) i.v. Animals were euthanized at week 7. (**a**) Tumor growth was monitored daily. Results are means + SEM (n = 3). * *p* < 0.001: S/Enox-Pa/DTX or S/DTX vs. ctrl (week 4–7 for S/DTX; weeks 2–7 for S/Enox-Pa/DTX); ° *p* < 0.001: S/Enox-Pa/DTX vs. DTX at the same dosage (weeks 2–7); # *p* < 0.01: S/Enox-Pa/DTX vs. S/DTX at the same dosage (weeks 4–7). (**b**) Representative photos of excised tumors. (**c**) Volumes of excised tumors. Results are means + SEM (n = 3). * *p* < 0.05: S/Enox-Pa/DTX or S/DTX vs. ctrl; ° *p* < 0.01: S/Enox-Pa/DTX vs. DTX at the same dosage; # *p* < 0.05: S/Enox-Pa/DTX vs. S/DTX at the same dosage. (**d**) Representative hematoxylin-eosin (HE) and Ki67 staining in each group of treatments. For each experimental condition a minimum of five fields were examined. Ocular: 10×; objective: 10×. Scale bar: 50 µm.

**Table 1 pharmaceutics-14-00292-t001:** Size (nm), PI, and ζ potential (mV) of SEDDS formulations in PBS (20 mM, pH 7.4). Values are reported as mean ± SD.

Formulation	Mean Diameter (nm ± SD)	PI (Mean ± SD)	ζ-Potential (mV ± SD)
S	109.2 ± 1.5	0.21 ± 0.02	−8.8 ± 0.7
S/DTX	113.9 ± 1.6	0.24 ± 0.01	−11.2 ± 1.0
S/Enox-Pa	115.3 ± 1.3	0.23 ± 0.01	−14.6 ± 0.8
S/Enox-Pa/DTX	144.8 ± 3.7	0.24 ± 0.05	−15.0 ± 1.6

**Table 2 pharmaceutics-14-00292-t002:** Size and PI of SEDDS formulations following incubation in serum albumin (1% *w*/*v*) or in plasma (1:100 dilution in phosphate buffer) at time zero and after 4 h at 37 °C. Values are reported as mean ± SD (n = 3).

BSA 37 °C				
Formulation	Mean Diameter (nm ± SD)	Mean Diameter (nm ± SD)	PI (Mean ± SD)	PI (Mean ± SD)
	Time 0	4 h	Time 0	4 h
S	127.9 ± 8.2	125.5 ± 1.3	0.21 ± 0.04	0.21 ± 0.04
S/DTX	114.4 ± 0.5	124.8 ± 0.2	0.19 ± 0.07	0.21 ± 0.01
S/Enox-Pa	135.1 ± 4.1	135.2 ± 8.2	0.27 ± 0.04	0.23 ± 0.02
S/Enox-Pa/DTX	158.1 ± 2.2	159.6 ± 7.1	0.44 ± 0.16	0.49 ± 0.04
**Human Plasma 37 °C**				
**Formulation**	**Mean Diameter (nm ± SD)**	**Mean Diameter (nm ± SD)**	**PI** **(Mean ± SD)**	**PI** **(Mean ± SD)**
	Time 0	4 h	Time 0	4 h
S	114.7 ± 0.7	125.5 ± 0.6	0.16 ± 0.03	0.17 ± 0.01
S/DTX	118.1 ± 0.4	127.9 ± 1.8	0.16 ± 0.01	0.17 ± 0.01
S/Enox-Pa	113.2 ± 1.5	121.7 ± 5.3	0.25 ± 0.04	0.16 ± 0.04
S/Enox-Pa/DTX	146.7 ± 4.4	151.4 ± 10.6	0.41 ± 0.23	0.41 ± 0.23

**Table 3 pharmaceutics-14-00292-t003:** Hemolysis (%) of SEDDS formulations after incubation in human blood (samples diluted 1:10 with NaCl 0.9% *w*/*v*) at 37 °C. Indicated values are mean ± SD (n = 3).

Formulation	Hemolysis (%)
S	1.6 ± 0.1
S/DTX	2.3 ± 0.2
S/Enox-Pa	1.6 ± 0.0
S/Enox-Pa/DTX	1.6 ± 0.1

**Table 4 pharmaceutics-14-00292-t004:** Size and PI of formulations with and without Docetaxel in NaCl (0.9% *w/v*) at time 0 and after filtration through a cellulose acetate filter (pore size: 0.2 µm). Indicated values are mean ± SD (n = 3).

	Mean Diameter (nm ± SD)		PI (Mean ± SD)	
Formulation	Before Filtration	After Filtration	Before Filtration	After Filtration
S	101.62 ± 5.48	102.40 ± 0.19	0.18 ± 0.02	0.19 ± 0.03
S/DTX	115.18 ± 0.98	116.37 ± 0.27	0.17 ± 0.01	0.13 ± 0.01
S/Enox-Pa	116.10 ± 0.24	112.35 ± 2.57	0.23 ± 0.02	0.27 ± 0.03
S/Enox-Pa/DTX	141.13 ± 2.85	142.70 ± 4.22	0.23 ± 0.05	0.28 ± 0.15

## Data Availability

The data presented in this study are available on request from the corresponding author.

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
