# Peer review of "Targeted Self-Emulsifying Drug Delivery Systems to Restore Docetaxel Sensitivity in Resistant Tumors"

_pharmaceutics, 2022, doi:10.3390/pharmaceutics14020292_

Round 1

Reviewer 1 Report

Campani et al. in this excellent report present their findings on the development of an enoxaparin coated self-emulsifying delivery system (SEDDS) which shows superior delivery of docetaxel. While similar studies have been previously reported, proving their applicability in a clinical setting for oral delivery, the application of self-emulsifying delivery systems for intravenous applications is underexplored. This study is important due to its attempts to address this ability and also incorporate the use of low molecular weight heparin (enoxaparin in this case) which contributes to the most common drawback in these therapies – drug resistance. Interestingly, upon a thorough in vitro screen, the authors note that cells with higher levels of e fibroblast growth factor receptor 1 demonstrated enhanced inhibition of cell growth upon treatment with enox-palmitoyl coated SEDDS. This contributes to the overall increase in uptake. Moreover, these treatments were able to restore the sensitivity of cells expressing MRP1 and BCRP receptors to docetaxel via their inhibition.  In support of the hypothesis, the formulation did not show any significant toxicity and also restored the efficacy of docetaxel in resistant NSCLC. Their experiential results conveyed the hypothesis in detail. All the data looks promising and I would recommend the publication of this manuscript in pharmaceutics in its current form.

Reviewer 2 Report

In the proposed paper, titled “Targeted self-emulsifying drug delivery systems to restore Docetaxel sensitivity in resistant tumors” the authors V. Campanini et al. proposed a self-emulsifying drug delivery systems (briefly, SEDDS) loaded with enoxaparin (Enox-Pa) and a chemotherapy drug, Docetaxel (DTX) as a promising strategy to deliver DTX in resistant tumours. As first, the authors described the synthesis of SEED and their chemical and physical characterization, in term of mean diameter, Z-potential, PI by means of DLS. Moreover, the authors performed stability studies in biological media and hemolysis assay. Finally, the authors described the preliminary in vitro  experiments (performed on numerous cells type) and in vivo studies. The manuscript might be very interesting and relevant to the pharmaceutics, nanobiotech, nanomedicine and, more in general, for medical community, however, it requires several modification before publication on Pharmaceutics. There are some relevant formal and technical issues that must be addressed.

  1. The article is well written, no relevant grammatical errors are found. Anyway, the article is not easy to read for many reasons. As first, the authors should revise the paper providing in the introduction, a description of the research background/state of art (the DXT  pharmacokinetics, for instance). I am conscious that these aspects are not the focus of this research work but, in my opinion, it would be useful to facilitate the understanding of the paper and increase the interest in the reader.

  1. The article is full of abbreviations and acronyms. Usually, the use of the acronyms should facilitate the reader but, not in this case. In fact, a list of abbreviations must be inserted at the beginning of the manuscript.

  1. Why did you choose the Breast and lung cancer cells lines for the in vitro experiments? This should be clarified in the manuscript.

  1. The manuscript section from the line n° 284 to the one 298 (after Statistical analysis section) must be deleted because this text is part of the template.

  1. The table 2 (pag. 9 of 19) should be revised. The errors column should be reported close to the column data which refers to. The mean size of S/Enox-Pa/DTX, both incubated in BSA and human plasma, decreases after 4 hours from about 158 to 139 nm and from 151 to 146 nm, respectively. How the authors explained this finding? The samples were submitted to any treatment (such as, vortex, US)?

  1. Have the authors performed tests to verify that the SEDDS solutions with were correctly sterile? Why did the authors avoid other physical sterilization techniques, such as UV?

  1. It would be useful for the authors to repot the confocal images of the other incubation time points (perhaps, in an panel in the supplementary materials). In this way, it would be possible to appreciate morphological changes of the cells induced by the cells incubation with the nano drugs. In addition, the quality of the confocal image 2 reported by the authors are not so good; it is very difficult to see the cells.

  1. Have the authors already identified possible collateral effects of the formulation?

  1. I really appreciated the in vitro and in vivo experiments that are, in my opinion, the strong point of this experimental work. However, these findings are not well highlighted in the introductions. Authors should find a better way to highlight them.
  2. All the references number are reported after the dots. It is an unusual way to report reference. The authors must report them before the dot: for instance: "..expression of efflux transporters [2]." instead of  "..expression of efflux transporters.[2]" 

Reviewer 3 Report

The submitted manuscript is original and in the scoop of the journal. In addition, the manuscript is well illustrated.

There are several questions to paper.

How much DTX is loaded into SEDDS? How is the utilization rate calculated? Does drug design affect loading in SEDDS? Can I use another medication instead of DTX (doxorubic, ibuprofen, naproxen, idarubicin, etc.)?

Under what conditions does DTH release occur? Is the entire drug released? What method was used to confirm this?

The synthesis of enoxaparin-palmitoyl conjugate is very similar to the synthesis of solid lipid nanoparticles. What structure do these kongats have? Is it a Micelle, liposome, solid lipid nanoparticle or others? How was the conjugate structure confirmed? What is their shape? Does the form affect the effectiveness of inhibition of MRP1 and BCRP?

Reviewer 4 Report

The manuscript is very well written; however, there are some mistakes in the English grammar and there should be some revisions.
